# Improved estimation of the bulk ice crystal fabric asymmetry from polarimetric phase co-registration

Ole Zeising[1], Tamara Annina Gerber[2], Olaf Eisen[1,3], M. Reza Ershadi[4], Nicolas Stoll[1,3], Ilka Weikusat[1,4], and Angelika Humbert[1,3]

[1]Alfred-Wegener-Institut Helmholtz-Zentrum für Polar- und Meeresforschung, Bremerhaven, Germany
[2]Section for the Physics of Ice, Climate and Earth, The Niels Bohr Institute, University of Copenhagen, Copenhagen, Denmark
[3]Department of Geosciences, University of Bremen, Bremen, Germany
[4]Department of Geosciences, Tübingen University, Tübingen, Germany

**Correspondence:** Ole Zeising (ole.zeising@awi.de)

**Abstract.** The bulk crystal orientation in ice influences the flow of glaciers and ice streams. The ice $c$-axes fabric is most reliably derived from ice cores. Because these are sparse, the spatial and vertical distribution of the fabric in the Greenland and Antarctic ice sheets is largely unknown. In recent years, methods have been developed to determine fabric characteristics from polarimetric radar measurements. The aim of this paper is to present an improved method to infer the horizontal fabric asymmetry by precisely determining the travel-time difference using co-polarised phase-sensitive radar data. We applied this method to six radar measurements from the EastGRIP drill site on Greenland's largest ice stream to give a proof-of-concept by comparing the results with the horizontal asymmetry of the bulk crystal anisotropy derived from the ice core. This comparison shows an excellent agreement, which is a large improvement compared to previously used methods. Our approach is particularly useful for determining the vertical profile of the fabric asymmetry in higher resolution and over larger depths than was achievable with previous methods, especially in regions with strong asymmetry.

## 1 Introduction

The distribution of the crystallographic-axis ($c$-axis) orientation fabric (henceforth *fabric*) in glaciers and ice sheets is a result of ice deformation history that can influence present-day ice-flow dynamics (Alley, 1988; Faria et al., 2014). Due to the mechanical anisotropy of ice crystals, the bulk viscosity is a directional quantity, spanning several orders of magnitude depending on the orientation of stresses with respect to the fabric type and orientation (Cuffey and Paterson, 2010). While some ice-flow models already account for fabric evolution and/or its effect on ice flow (e.g. Thorsteinsson, 2002; Gillet-Chaulet et al., 2006; Seddik et al., 2008; Martín et al., 2009), the validation of these models is challenged by the lack of in-situ data.

Most reliably, the crystal fabric of ice can be determined from the analysis of ice core thin sections (e.g. Thorsteinsson et al., 1997; Azuma et al., 1999; Wang et al., 2002; Montagnat et al., 2014; Weikusat et al., 2017). Since deep ice cores are sparse in Greenland and Antarctica, and often restricted to domes with rather undisturbed stratigraphy, little is known about the spatial distribution of crystal fabric anisotropy of ice sheets. It is therefore of great importance to use other methods in order to infer

the spatial and vertical distribution of the fabric asymmetry, e.g. for improving ice-flow models and determining past flow and deformation.

Ice crystals are uniaxially birefringent (Hargreaves, 1978). This means that ice crystals are dielectrically anisotropic due to crystal anisotropy and thus allow the horizontal fabric asymmetry to be determined from polarimetric radar surveys (e.g. Fujita et al., 2006; Drews et al., 2012; Leinss et al., 2016; Brisbourne et al., 2019; Jordan et al., 2019, 2020; Young et al., 2021a, b; Ershadi et al., 2022; Jordan et al., 2022; Gerber et al., 2022), with certain limitations (Rathmann et al., 2022). Since polarimetric radar measurements are easier to conduct than ice core analyses, they enable a greater spatial coverage and thus offer the opportunity to examine the distribution of fabric asymmetry. For vertically propagating radio waves, the relevant dielectric anisotropy is the difference between the bulk horizontal permittivities (Fujita et al., 2000). One way of inferring the horizontal fabric asymmetry is based on a polarimetric coherence method (Dall, 2010), which refers to the strength of the phase correlation between orthogonal polarisations. This method has recently been applied to polarimetric radar data and compared with the fabric asymmetry from the NEEM ice core in Greenland (Jordan et al., 2019), WAIS divide ice core in West Antarctica (Young et al., 2021a) or the EDML and EDC ice cores in East Antarctica, respectively (Ershadi et al., 2022). However, this method has some limitations (Leinss et al., 2016). Most importantly, the method can either only be used where the asymmetry of the fabric is weak or otherwise its application is limited to shallow depth (Jordan et al., 2022), which we discuss later in detail.

In this study, we infer the horizontal asymmetry of the bulk crystal fabric at the East Greenland Ice Core Project (EastGRIP) drill site from co-polarised phase-sensitive radar measurements by using a new, improved coherence method. This method differs from previously used analysis schemes and has the advantage that the asymmetry can be determined with much higher vertical resolution and, regardless of its strength, up to the onset of the noise level. We present a proof-of-concept by comparing the derived horizontal fabric asymmetry with those from the ice core analysis. A glaciological interpretation of the detected fabric asymmetry regarding the flow dynamics in the region of the EastGRIP drill site is part of a larger study by Gerber et al. (2022) and we refer to their study for ice-dynamical interpretations.

## 2  Data

In order to investigate ice flow dynamics of Greenland's largest ice stream, the Northeast Greenland Ice Stream (NEGIS), an ice core is being drilled through the $\sim 2668\,\mathrm{m}$ thick ice (Zeising and Humbert, 2021) as part of EastGRIP. In the vicinity of the EastGRIP drill site (75.63 °N, 35.99 °W in 2019), we performed polarimetric measurements with a phase-sensitive radio echo sounder (pRES; Brennan et al. (2014); Nicholls et al. (2015)) in 2019: within the drill trench next to the core location (*CL*; $\sim 10\,\mathrm{m}$ apart) and at five sites (called *GRID*) within $20 \times 20\,\mathrm{m}^2$ approximately $360\,\mathrm{m}$ from the drill site (Fig. 1). These five sites are labelled depending on their cardinal direction (N, E, S and W) compared to the centre point (C). The pRES is a ground-based nadir looking Frequency-Modulated Continuous-Wave (FMCW) radar, which allows to determine vertical displacements of reflections within firn and ice from repeated measurements with a high precision of $\sim 1\,\mathrm{mm}$. While the pRES is mainly operated to derive basal melt rates (e.g. Marsh et al., 2016; Stewart et al., 2019; Zeising et al., 2022), it also can

be used to estimate the ice fabric from polarimetric measurements (Brisbourne et al., 2019; Jordan et al., 2020; Young et al., 2021a; Ershadi et al., 2022; Jordan et al., 2022).

A polarimetric pRES measurement consists of several measurements with different antenna orientations. The pRES transmits linearly polarised electromagnetic waves via the transmitting skeleton slot antenna and records the reflected signals in one direction with another antenna. This allows co-polarised measurements to be made in which the two antennas are oriented

in the same direction. While in a $hh$ measurement the direction of polarization points to each other, in a $vv$ measurement it is perpendicular to the $hh$ measurement (Fig. 1c). Recent studies used quad-polarised acquisitions which additionally include $hv$ and $vh$ measurements, where the polarisation direction of the transmitting and receiving antenna is rotated by 90° (e.g. Brisbourne et al., 2019; Young et al., 2021a; Ershadi et al., 2022; Jordan et al., 2022). However, this study focuses on co-polarised measurements.

We aligned the antennas at an arbitrary azimuthal angle of roughly 258° (at CL) and 168° (at GRID) clockwise to magnetic North (283° and 193° true North), respectively. The ice flow direction at EastGRIP is roughly 58° magnetic North. We performed multi-polarised measurements by rotating each antenna separately horizontally clockwise in 22.5° steps up to 157.5°. Here, we only considered the co-polarised measurements taken roughly in ice flow direction ($hh$; 55° to magnetic North) and perpendicular to ice flow ($vv$; 145° to magnetic North). During each measurement, the pRES transmitted a sequence of

chirps by linearly increasing the transmitted frequency from 200 to 400 MHz within 1 s for each chirp. In order to achieve a higher signal-to-noise ratio, the measurement at CL contained 250 chirps and those of the GRID contained 100 chirps per measurement.

## 3  Methods

### 3.1  Fabric anisotropy from ice core analysis

Every 5–15 m of depth of the ice core a 55 cm long section was analysed for fabric data. Details of the sample preparation, data acquisition and processing are given in Stoll et al. (2021). The grain size weighted orientation of the measured $c$-axes can be represented by the second-order orientation tensor. Its normalised eigenvalues

$$\lambda_1 + \lambda_2 + \lambda_3 = 1 \ \text{ and } \ \lambda_1 \leq \lambda_2 \leq \lambda_3 \tag{1}$$

quantifies the strength of the three principal fabric ($c$-axis) directions. In order to determine the fabric asymmetry, we averaged

those eigenvalues from all samples of each section and calculated the difference between the eigenvalues ($\lambda_2 - \lambda_1$ and $\lambda_3 - \lambda_1$).

### 3.2  Horizontal fabric asymmetry from radar measurements

If two electromagnetic waves, whose polarisation in $x'$ and $y'$ are perpendicular to each other, propagate through an anisotropic medium, their depth-averaged propagation velocities $\overline{v}_{x'}$ and $\overline{v}_{y'}$ differ due to the horizontal dielectric anisotropy (Hargreaves,

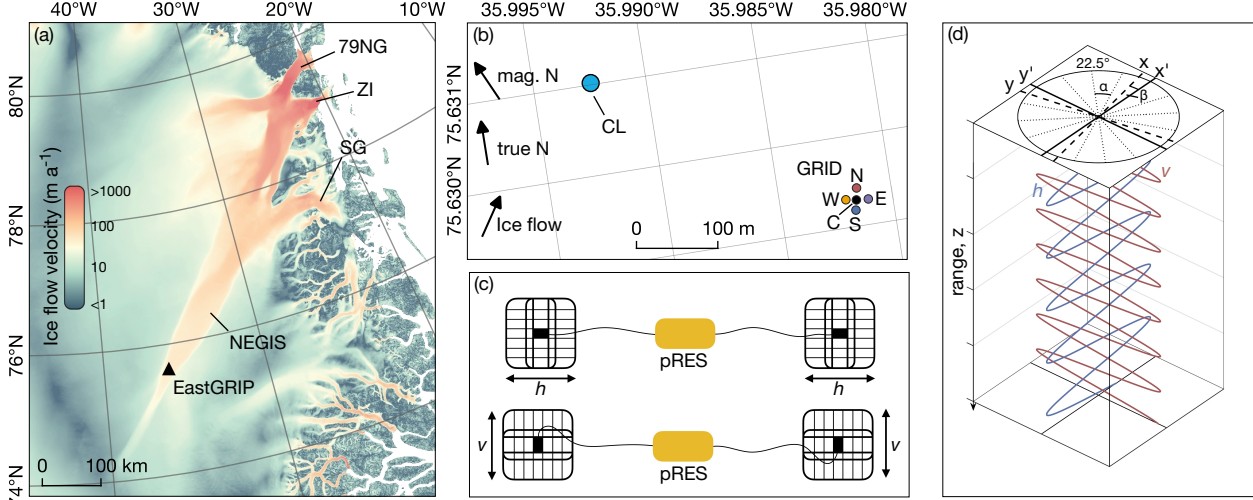

**Figure 1.** Location and orientation of polarimetric pRES measurements. (a) Surface ice flow velocity of the Greenland Ice Sheet (Joughin et al., 2016, 2018), showing the three major outlet glaciers of the Northeast Greenland Ice Stream (NEGIS): Nioghalvfjerdsbrae (79 N Glacier, 79NG), Zachariæ Isstrøm (ZI) and Storstrømmen Glacier (SG). The location of the EastGRIP drill site is denoted by the black triangle. (b) Location of polarimetric pRES measurements at CL and at GRID. Arrows shown direction of magnetic North, true North and ice flow direction. (c) Sketch of a polarimetric pRES measurement with $hh$ and a $vv$ antenna orientation. (d) Sketch of propagating waves with polarisations in $x'$ ($h$) and $y'$ ($v$) direction (solid line) in the x-y coordinate system (dashed line). Dotted lines show the (unused) multi-polarised measurements, separated by $\alpha = 22.5°$. Ice flow is in $x$ direction with an angular offset of $\beta$ to the $hh$-measurement in $x'$ direction.

1978):

$$\overline{v}_{x'}(z) = \frac{c_0}{\sqrt{\overline{\varepsilon}_{x'}(z)}} = \frac{2z}{t_{x'}(z)}, \tag{2}$$

$$\overline{v}_{y'}(z) = \frac{c_0}{\sqrt{\overline{\varepsilon}_{y'}(z)}} = \frac{2z}{t_{y'}(z)} \tag{3}$$

where $c_0$ is the speed of light in vacuum, $t_{x'}$, $t_{y'}$ are the two-way travel times to a reflector at depth $z$ and $\overline{\varepsilon}_{x'}$, $\overline{\varepsilon}_{y'}$ are the permittivities averaged over the whole depth in the corresponding polarization directions $x'$, $y'$. The resulting difference in two-way travel-time $\Delta t_{x'y'}$ of a backscatter from a reflector at depth $z$ is

$$\Delta t_{x'y'}(z) = t_{y'}(z) - t_{x'}(z) = \frac{2\left(\sqrt{\overline{\varepsilon}_{y'}(z)} - \sqrt{\overline{\varepsilon}_{x'}(z)}\right)}{c_0} z, \tag{4}$$

and thus the vertical profile of the depth-averaged permittivities are

$$\overline{\varepsilon}_{x'}(z) = \frac{c_0^2}{4}\left(\frac{t_{x'}(z)}{z}\right)^2, \tag{5}$$

$$\overline{\varepsilon}_{y'}(z) = \frac{c_0^2}{4}\left(\frac{t_{x'}(z) + \Delta t_{x'y'}(z)}{z}\right)^2. \tag{6}$$

These dielectric permittivities are the average values over the entire depth from the surface to the depth $z$. In order to calculate the vertical profile of the horizontal dielectric anisotropy $\Delta\varepsilon_{x'y'} = \varepsilon_{y'} - \varepsilon_{x'}$, the local change in two-way travel time $\delta t_{x'}$, $\delta t_{y'}$ for a given infinitesimal depth window $\delta z$, needs to be taken into account:

$$\varepsilon_{x'}(z) = \frac{c_0^2}{4}\left(\frac{\delta t_{x'}(z)}{\delta z}\right)^2, \tag{7}$$

$$\varepsilon_{y'}(z) = \frac{c_0^2}{4}\left(\frac{\delta(t_{x'}(z) + \Delta t_{x'y'}(z))}{\delta z}\right)^2. \tag{8}$$

If it is assumed that the ice crystals are an effective medium at ice penetrating frequencies, the bulk horizontal dielectric anisotropy for the two polarisations in $x'$ and $y'$ direction, $\Delta\varepsilon_{x'y'}$, is a function of the horizontal fabric asymmetry $\Delta\lambda_{x'y'} = \lambda_{y'} - \lambda_{x'}$ and of the dielectric anisotropy of an ice crystal $\Delta\varepsilon'$:

$$\Delta\varepsilon_{x'y'}(z) = \varepsilon_{y'}(z) - \varepsilon_{x'}(z) = \Delta\varepsilon'(\lambda_{y'}(z) - \lambda_{x'}(z)) = \Delta\varepsilon'\Delta\lambda_{x'y'}(z) \tag{9}$$

(Fujita et al., 2006; Jordan et al., 2019). This assumes that the wavelength is much longer than the average grain size, which is the case for the frequency range from 200 to 400 MHz and the corresponding wavelengths from 0.42 to 0.84 m. Matsuoka et al. (1997) found $\Delta\varepsilon' = 0.034$ for ice-penetrating radar frequencies. Finally, the horizontal fabric asymmetry $\Delta\lambda_{x'y'}$ at depth $z$ is given by

$$\Delta\lambda_{x'y'}(z) = \lambda_{y'}(z) - \lambda_{x'}(z) = \frac{\Delta\varepsilon_{x'y'}(z)}{\Delta\varepsilon'}. \tag{10}$$

Thus, the bulk dielectric anisotropy $\Delta\varepsilon_{x'y'}$, and based on this, the horizontal fabric asymmetry $\Delta\lambda_{x'y'}$ can be determined from the difference in the two-way travel time $\Delta t_{x'y'}$. The vertical resolution of $\Delta\lambda_{x'y'}$ depends on the precise determination of $\Delta t_{x'y'}$, which used to be a problem for previous radar systems that did not have the required resolution in the time domain. This is the main advantage of the in-depth analysis of the phase which is why polarimetric pRES measurements offer the chance to investigate the horizontal fabric asymmetry in the ice.

### 3.3 Phase-sensitive radar data analysis

For data processing, we followed Brennan et al. (2014) and Stewart et al. (2019) in order to get the complex valued signals $s_{hh}$ and $s_{vv}$ (subscripts indicate the transmitted and received polarisation) as a function of two-way travel time with the amplitude $|s_{hh}|$ and its phase. We convert $s$ from time $t$ to depth $z$ domain by using dielectric permittivities derived from dielectric profiling (DEP) of the EastGRIP ice core by Mojtabavi et al. (2020).

The method we apply to compute the travel-time difference $\Delta t_{x'y'}$ is based on a cross-correlation of the co-polarised measurements. The same method is widely used to estimate vertical displacements for strain analysis from repeated pRES measurements as shown by e.g. Jenkins et al. (2006), Gillet-Chaulet et al. (2011), Stewart et al. (2019) and Zeising and Humbert (2021). We divided $s_{hh}$ into segments of 12 m depth overlapping by 9 m and calculated for each the complex coherence

$$c_{hhvv}(z,l) = \frac{\sum_{j=i_n}^{i_n+N} s_{hh}(j)s_{vv}^*(j+l)}{\sqrt{\sum_{j=i_n}^{i_n+N}|s_{hh}(j)|^2}\sqrt{\sum_{j=i_n}^{i_n+N}|s_{vv}(j+l)|^2}}, \tag{11}$$

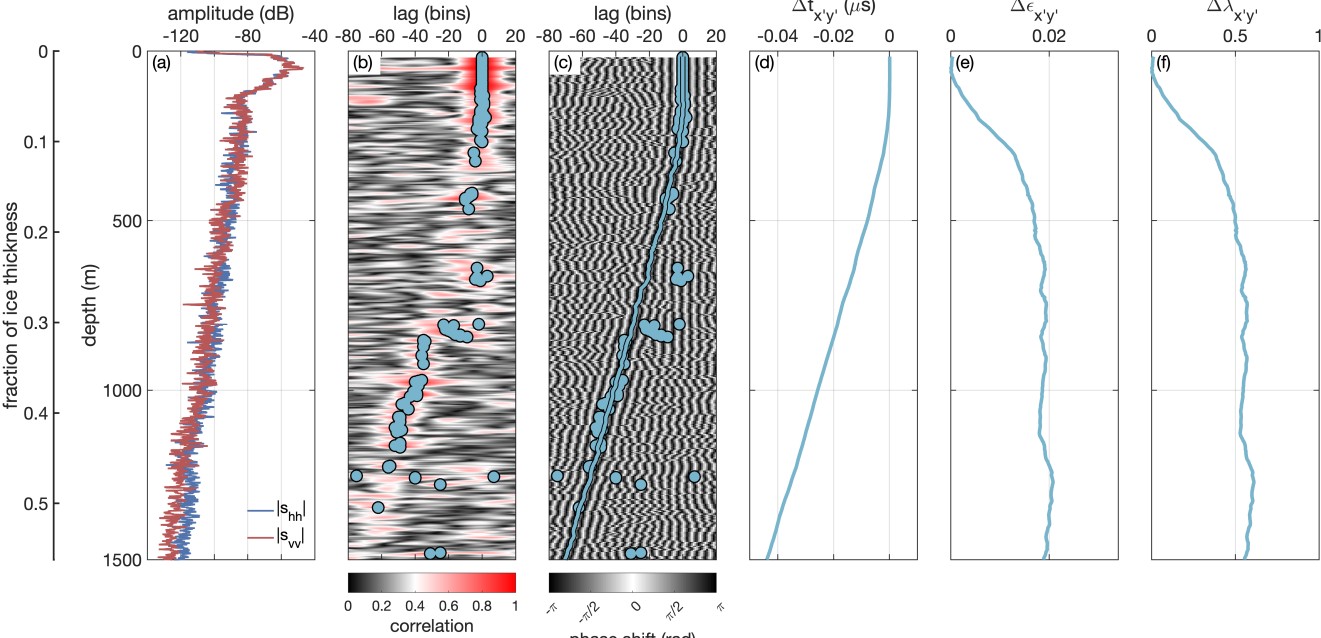

**Figure 2.** Analysis of the horizontal fabric asymmetry from polarimetric pRES measurements at the location CL (Fig. 1) next to the EastGRIP ice core. (a) Magnitude profiles of $s_{hh}$ (blue line) and $s_{vv}$ (red line) as a function of depth. (b) Cross-correlation $|c_{hhvv}|$ of $s_{hh}$ and $s_{vv}$ as a function of lag and depth. Blue dots mark the lag of best correlation for each segment exceeding a correlation of 0.65. (c) Coherence phase shift $\phi_{hhvv}$ as a function of lag and depth. The blue dots are the same as in (b). The blue line marks the tracked minimum phase shift. (d) Difference in two-way travel time between both measurements at the same depth after smoothing with a $100\,\mathrm{m}$ moving average filter. (e) Difference in horizontal dielectric anisotropy $\Delta\varepsilon_{x'y'}$. (f) Difference in horizontal eigenvalues $\Delta\lambda_{x'y'}$.

where $i_n$ is the lower time-bin index of the segment, $N$ the number of bins in the segment, $l$ the range-bin offset (lag) and $^*$
indicates the complex conjugate (Stewart et al., 2019). While the magnitude of the complex coherence $|c_{hhvv}|$ is the correlation
between $s_{hh}$ and $s_{vv}$, the argument is the coherence phase $\phi_{hhvv} = \arg(c_{hhvv})$ (Jordan et al., 2019).

   Our *polarimetric cross-correlation* approach differs from the *coherence* method from Dall (2010) that was used by Jordan
et al. (2019, 2020, 2022), Young et al. (2021a) and Ershadi et al. (2022). In their applications, the range-bin offset was set
to zero ($l = 0$). Thus, these studies interpreted the $hhvv$ coherence phase gradient of the same two-way travel-time. In this
study, we are analysing the travel-time difference of the same reflector that we determine from a cross-correlation approach.
We co-register the phase of $s_{hh}$ and $s_{vv}$ for every segment by shifting $s_{vv}$ by a number of integer bin offsets $l$ (see eq. 11). We
identified the correct $l$ of each segment by following the minimum phase difference from the surface downwards, indicated by
high correlation values (Fig. 2b,c).

Next, we compute the travel-time difference $\Delta t_{x'y'}$ (Fig. 2d) for each segment based on the selected lag $l$ and the corresponding coherent phase $\phi_{hhvv}$ (see Brennan et al., 2014):

$$\Delta t_{x'y'}(z) = \frac{l(z)}{Bp} + \frac{\phi_{hhvv}(l,z)}{2\pi f_c}. \tag{12}$$

The first term on the right side is the coarse time-bin offset with $1/Bp$ being the time-bin spacing ($B = 200\,\mathrm{MHz}$ is the bandwidth and $p = 8$ is a 'padding factor' that reduces the range-bin spacing). The second term is the fine offset derived from the coherent phase of the centre frequency of $f_c = 300\,\mathrm{MHz}$.

Since the travel-time difference is cumulative, we calculated the mean vertical change of the two-way travel times, $\delta t_{x'}/\delta z$ and $\delta(t_{x'} + \Delta t_{x'y'})/\delta z$, from a $200\,\mathrm{m}$ moving window after smoothing $\Delta t_{x'y'}$ with a $100\,\mathrm{m}$ moving average filter. Between the surface and $100\,\mathrm{m}$ depth, we changed the method to use a smaller, adaptive moving window that increases with depth. Finally, we compute the dielectric anisotropy $\Delta\varepsilon_{x'y'}$ from Eqs. 8 and 9 (Fig. 2e) and the horizontal fabric asymmetry $\Delta\lambda_{x'y'}$ from Eq. 10 (Fig. 2f).

## 4 Results

The horizontal fabric asymmetry from the polarimetric cross-correlation analysis at all measurement locations show the same vertical distribution with only minor differences (Fig. 3a). They indicate a rapid increase of $\Delta\lambda_{x'y'}$ from $0.06$ to $0.4$ within the first $12\%$ of the ice thickness between $125$ and $320\,\mathrm{m}$ depth. This is followed by a minor increase to $0.55$, reached at a depth of $550\,\mathrm{m}$ ($20\%$ of the ice thickness). Between this depth and $1400\,\mathrm{m}$, the horizontal anisotropy remains at high level and varies between $0.52$ and $0.62$. Below the depth of $1400\,\mathrm{m}$, a low signal-to-noise ratio prevented the analysis of the horizontal fabric asymmetry. This depth corresponds to $52\%$ of the ice thickness.

In order to demonstrate the improvement over the previous coherence method, we also applied the method from Young et al. (2021a), which is based on the work of Jordan et al. (2019, 2020). The results show the same vertical profile only between $100$ and $260\,\mathrm{m}$. Below, the horizontal asymmetry drops to near zero and differs strongly from the result of the new cross-correlation method.

The pRES-derived vertical distribution matches the distribution of the difference of the weighted horizontal eigenvalues from the EastGRIP ice core analysis nearly perfectly (Fig. 3b).

While the differences of the first two eigenvalues ($\lambda_2 - \lambda_1$) show the same rapid increase between $125$ and $250\,\mathrm{m}$ depth, below, it is $\lambda_3 - \lambda_1$ that is of the same size than the pRES-derived values. This indicates that one of the horizontal eigenvalues becomes the largest value ($\lambda_3$ by definition) at a depth of $250\,\mathrm{m}$ and thus $\lambda_3$ switches from the vertical to one horizontal axis. However, since $\Delta\lambda_{x'y'}$ exceeds $0.5$, it is obvious that a horizontal eigenvalue is the largest since the $\lambda_2 - \lambda_1$ is always $\leq 0.5$.

## 5 Discussion

Our polarimetric cross-correlation method allows to resolve the travel-time difference of the co-polarised waves with sub-nanoseconds resolution. On this basis, the vertical profile of the horizontal dielectric anisotropy as well as the bulk crystal fabric

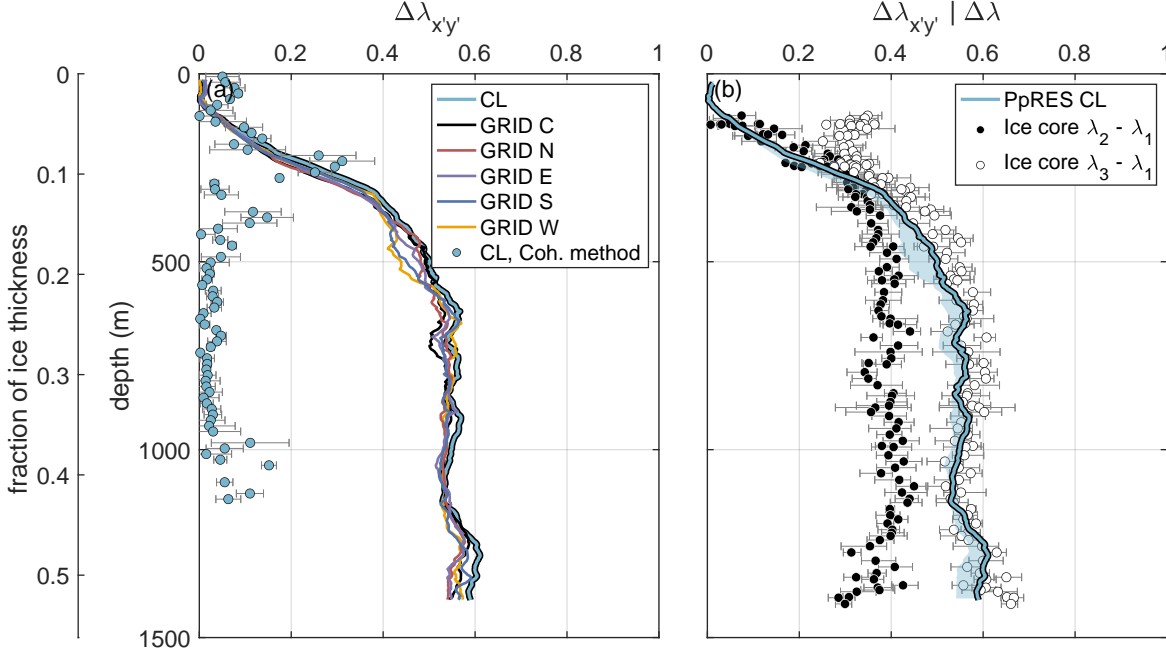

**Figure 3.** Comparison of horizontal fabric asymmetry $\Delta\lambda$ determined from different measurements and analysing methods. (a) Fabric asymmetry determined from cross-correlation analysis (lines) of pRES measurements at CL (light blue line) and at the $20 \times 20\,\mathrm{m}$ GRID outside drill site as well as from the previous coherence method (Young et al., 2021a) at CL (light blue dots). (b) Fabric asymmetry determined from cross-correlation analysis (lines) of pRES measurements at CL (light blue line) and from weighted horizontal eigenvalues from EastGRIP ice core (black and white dots). The blue shaded area in (b) marks the range of the polarimetric pRES-derived asymmetry from the measurements in the GRID and at CL.

asymmetry can be determined. Despite the high range resolution, the scatter of $\Delta t_{x'y'}$ caused by the uncertainty prevented a
165 determination of the small-scale gradient of the travel-time difference. Thus, the derived horizontal anisotropy only represents a coarse distribution. The horizontal fabric asymmetry derived from the polarimetric cross-correlation of the pRES measurements and the difference of the weighted horizontal eigenvalues from the ice core analysis ($\lambda_2 - \lambda_1$ between 120 and $250\,\mathrm{m}$ and $\lambda_3 - \lambda_1$ between 250 and $1400\,\mathrm{m}$) show excellent agreement with a root-mean-square difference of the result of both methods of only $0.03$, which corresponds to the uncertainty of the ice core analysis. However, the root-mean-square value of the difference of
170 the unweighted horizontal eigenvalue is $0.06$ and thus higher, which is a result compatible to analyses of seismic waves by Kerch et al. (2018).

The determination of the horizontal asymmetry is not possible for every azimuthal angle. The azimuth angle of the antenna has to match the alignment of the orientation of the ice fabric principal axes sufficiently. If the direction of polarization is rotated $45°$ to the alignment of the principal axes, no anisotropy can be determined, as the propagation velocity is the same in $x'$ and $y'$
direction. The polarimetric pRES measurements at EastGRIP show that with an azimuthal rotation of the antennas with $22.5°$

increments up to 67.5°, a determination is possible in two of the four orientations and that the derived horizontal anisotropy is identical in both cases. However, a clear advantage of quad-polarised measurements is that they allow to reconstruct co-polarised data at a high angular resolution and additionally the determination of the fabric orientation (e.g. Brisbourne et al., 2019; Young et al., 2021a; Ershadi et al., 2022; Jordan et al., 2022). The presented cross-correlation method can also be applied to these reconstructed co-polarised data. Since only four measurements ($hh$, $vv$, $hv$ and $vh$) at one azimuthal angle are necessary to perform a quad-polarised acquisition, but eight for co-polarised measurement ($hh$ and $vv$) at four different azimuthal angles (0°, 22.5°, 45° and 67.5°), quad-polarised measurements should be preferred in the future.

The previously used coherence method estimates the fabric asymmetry by determining the phase gradient of the polarimetric phase difference. This is also possible for high coherence persisting over a few phase cycles (e.g., Young et al., 2021a). However, in case of a strongly developed fabric asymmetry and thus a rapid phase-cycling, the coherence is reduced over depth because the segments that are correlated do not completely overlap and therefore contain different scatterers (Leinss et al., 2016). At ice divides or domes with very little asymmetry, such as at NEEM (Jordan et al., 2019), WAIS divide (Young et al., 2021a) or EDC (Ershadi et al., 2022), the fabric asymmetry could successfully be determined using previous coherence method up to the onset of noise. However, in fast moving areas like the Rutford Ice Stream, Antarctica (Jordan et al., 2022) or NEGIS, Greenland (this study) rapid phase-cycling limits the application of the previous coherence method to a few hundred meters below the surface. With the improved polarimetric cross-correlation method, we overcome this limitation through co-registration, which allows to determine even strong horizontal fabric asymmetries to a much greater depth. Noise limits the evaluation of fabric asymmetry for deeper layers. At the EastGRIP drill site, this limit is about half the ice thickness of the ice with current systems. Determining the fabric for deeper layers from radar measurements, eventually over the whole ice sheet thickness, requires further reduction of the signal-to-noise ratio in more powerful phase-sensitive radar system that can perform co- or quad-polarized measurements. The applicability of the polarimetric cross-correlation method needs first to be demonstrated for such radar systems.

Ershadi et al. (2022) presented a method to estimate horizontal ice fabric anisotropy based on a non-linear inverse approach by using the coherence phase gradient and power anomaly. Here we tried to use this method on our data to compare the two methods directly. However, the ice fabric orientation in this area rotates several times at different depths of the ice column, which prevents the application of the previous method using the inverse approach. Therefore, the attempt for direct comparison was unsuccessful and is another reason why we regard our method as an improvement which goes beyond previous limits.

## 6 Conclusions

We presented a new method to infer the vertical profile of the horizontal fabric asymmetry from polarimetric phase-sensitive radar measurements. Our approach is based on a cross-correlation of co-polarised measurements to derive precisely travel-time differences caused by dielectric anisotropy. In contrast to previous methods, this polarimetric cross-correlation approach allows to analyse even strong horizontal fabric asymmetries at a much greater depth.

The remarkable agreement between the vertical profile of the horizontal fabric asymmetry obtained by our analyses of multiple polarimetric pRES measurements and the fabric measured in the EastGRIP ice core demonstrates the robustness and precision of our method.

In the future, the applicability of our polarimetric cross-correlation method to other radar systems should be tested, in particular to polarimetric airborne radar measurements. If successful, this would increase the spatial coverage of mapped crystal fabric and its variability than would be possible with pointwise polarimetric pRES measurements. Furthermore, it might allow the estimation of the fabric to greater depth. Such an application, which would yield the variation of the horizontal anisotropy along flow lines or across regions of fast flow, like ice streams, would significantly improve the understanding of the link between the stress state and crystal fabric evolution. This would allow us to decrease uncertainties of rheology, and thus improve estimates for response times of dynamically active glacial systems to external perturbations, e.g. from changing ocean conditions of tidewater glaciers.

*Code and data availability.* Raw data of the multi-polarised pRES measurements (https://doi.org/10.1594/PANGAEA.951267, Zeising and Humbert, 2022) and crystal c-axes of ice core samples are published at the World Data Center PANGAEA (https://doi.org/10.1594/PANGAEA.949248, Weikusat et al., 2022). The MATLAB code of the polarimetric cross-correlation method is published at Zenodo (https://doi.org/10.5281/zenodo.7577772, Zeising, 2023). The MATLAB code of the coherence method from Young et al. (2021a) is available at NERC EDS UK Polar Data Centre (https://doi.org/10.5285/BA1CAF7A-D4E0-4671-972A-E567A25CCD2C, Young and Dawson, 2021).

*Author contributions.* OZ and AH designed the study and performed the polarimetric radar measurements. OZ processed the data together with MRE and prepared the manuscript with contributions from all co-authors. OZ and TAG developed the method with support from OE. NS and IW prepared the ice core samples used for fabric analyses, performed the measurements, and processed and analysed the fabric data. All authors contributed to writing and editing the manuscript.

*Competing interests.* OE is an editor for TC but has not competing interests.

*Acknowledgements.* Data has been acquired at the EastGRIP camp that kindly hosted this activity as an associate project. EastGRIP is directed and organized by the Centre for Ice and Climate at the Niels Bohr Institute, University of Copenhagen. It is supported by funding agencies and institutions in Denmark (A. P. Møller Foundation, University of Copenhagen), USA (US National Science Foundation, Office of Polar Programs), Germany (Alfred Wegener Institute, Helmholtz Centre for Polar and Marine Research), Japan (National Institute of Polar Research and Arctic Challenge for Sustainability), Norway (University of Bergen and Trond Mohn Foundation), Switzerland (Swiss National Science Foundation), France (French Polar Institute Paul-Emile Victor, Institute for Geosciences and Environmental research), Canada

(University of Manitoba) and China (Chinese Academy of Sciences and Beijing Normal University). Nicolas Stoll gratefully acknowledges funding from the Helmholtz Junior Research group "The effect of deformation mechanisms on ice sheet dynamics" (VH-NG-802).

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
