# Peer review of "Improved estimation of the bulk ice crystal fabric asymmetry from polarimetric phase co-registration"

_The Cryosphere, 2022_

## Referee Comment (RC1)

**Review to Zeising et al. 2022, The Cryosphere**

**Summary**

Zeising et al. present a technical development using phase-sensitive radar polarimetry to estimate ice crystal orientation fabric near the EastGRIP ice core. The key point of novelty is to enhance the polarimetric coherence (a method used by previous radar polarimetry studies) by adjusting the range-bin offset between orthogonal polarizations. This allows them to obtain high coherence throughout the ice column, which is then used to infer two-way travel time differences, dielectric anisotropy, and azimuthal fabric anisotropy. They then compare with the ice core fabric eigenvalue differences, showing very good agreement down to ice depths ~ 1500 m.

I like the general concept, and enjoyed reading the paper. Whilst the coherence optimization/range-bin offset method is a relatively simple refinement from previous studies, it produces agreement between ice-core fabric data and the radar asymmetry estimates (probably the best I have seen to date). However, as it stands, I think the paper needs to better demonstrate how it has improved coherence and fabric estimation from methods used in previous studies. Additionally, fabric orientation information should be provided (both from the radar, and comparison with the ice core if it is available) as other similar radar polarimetry studies have all done this. I also do not agree with the physical interpretation related to the `half-wavelength limit' (regarding why previous applications of polarimetric coherence will be ineffective) and have given some counter arguments and suggestions to rephrase the discussion.

At < 200 lines and 3 figures the study is significantly shorter than a typical TC article, and closer to what I would expect for a `TC brief communication'. I see no issue with this, but potentially the editorial team and authors will want to change the format for the final publication.

Best regards,

Tom Jordan, Plymouth Marine Laboratory, UK

**Specific/major comments**

1. *Demonstration of improved coherence and fabric estimates over previous methods.*

A central weakness of the study is that it does not explicitly demonstrate the improvement of the coherence magnitude and fabric estimates over previous methods. A reader less familiar with the field will therefore be uncertain about the progress made in the paper.

I think this can be fixed relatively easily by showing:

> (i) A depth profile for |chvv| calculated *without* the range-bin offset, similar to the second column in Fig 2. My guess is that this will decay rapidly with depth, showing the coherence method in the paper to be more effective than previous.

> (ii) Adding fabric asymmetry estimates, where possible, using the previous approach: i.e. using vertical gradient of the co-polarized phase difference (no offset) employed by Dall 2010, Jordan et al. 2019, 2021, 2022, Young et al. 2021.

Related to this point, I would put a qualifying statement that the study can only be directly compared with methods that have used multi-polarized data (e.g. Jordan et al. 2019, 2020) rather than quad-pol approaches (e.g. Brisbourne et al. 2019, Young et al. 2021, Ershadi et al. 2021, Jordan et al. 2022). A consequence of using quad-pol is that it enables reconstruction of multi-polarization data at higher angular resolution, which gives a particular advantage to inferring fabric orientation.

**2. Fabric orientation information.**

In addition to azimuthal asymmetry, polarimetric radar enables estimation of the orientation of (assumed) horizontal eigenvectors. This can be done by comparing data with the polarimetric backscatter model (Fujita 2006, Jordan et al. 2019) and using sign of the coherence phase gradient (Dall 2010). Fabric orientation is very useful information, as it informs about past deformation, aswell as being key information for understanding the impact of fabric on anisotropic rheology.

I therefore find the current study incomplete, and I think it would be significantly strengthened, by including fabric orientation panels in Fig 3 (i.e. azimuthal angle of horizontal eigenvectors as a function of ice depth). I appreciate that the angular resolution will be limited as multi-polarization rather than quad-pol was used, but I still think this will be a nice inclusion.

Additionally, does the EastGRIP core have azimuthal fabric orientation to compare with? If it does, then this should be included, in the orientation plots.

**3. Accuracy of the discussion on the `half-wavelength limit'**

In their discussion, the authors focus on the `half-wavelength limit' (for the polarimetric phase difference) as an explanation why the polarimetric coherence method has previously been limited. e.g. *`Due to the ambiguities caused by phase wrapping, the previous methods which are based on the coherence phase gradient were limited to the derivation of phase shifts of a maximum of half a wavelength, (Line 61)'.* I don't agree with this interpretation, and I have given some counter arguments and suggestions for revision below.

First, I don't think previous methods based on phase gradient are limited by the phase wrapping. Jordan et al. 2019 used an identity (equation 23) to differentiate the phase. This approach (adapted from the InSAR literature) gets around the issue of phase wrapping, as the real and imaginary components of the coherence are continuous functions, enabling the derivative to be taken. Figure 5 and 6 from this paper illustrates that the phase gradient can be obtained at the phase discontinuities.

There are other examples in the literature that illustrate that the coherence magnitude, phase difference, and fabric estimates are not physically limited by the phase discontinuities and the proposed `half-wavelength limit'. Notably, Young et al. 2021 (Fig. 4) shows high coherence persisting over multiple (~4) phase cycles.

It therefore follows that `strong azimuthal asymmetry' (and rapid phase-cycling) should not be a singular limitation on the previous method. I think it is probably coincidence that the fabric in Jordan et

al. 2022 is often only obtained in the first phase cycle (line 171). This study proposed a degradation in the radar stratigraphy as the reason for the coherence drop-off with depth.

Despite discounting the `half-wavelength limit' interpretation, I do agree with the authors that their optimization should lead to higher coherence, and therefore improve the fabric estimates. This is because their method should act to better co-register the signal from a given radar layer. I think, if they better show the impact of the optimization on the coherence (following my comment 1), then they will be able to refocus the discussion around improved co-registration being the physical mechanism that improves the coherence and fabric estimates. (As an aside, I don't think it is strictly necessary for the reflection to occur from the same layer for each polarization. As long as the layers behave as flat, specular, reflectors, the original coherence method should still work to some degree).

Aswell as the discussion, lines 35-38 and line 186,will also need addressing regarding this point.

**Minor comments/typos**

Title - The MS title is quite generic (in effect all COF/radar polarimetry studies estimate horizontal asymmetry!). As the phase co-registration/optimization of the coherence is the key point of novelty, I recommended changing to something like: `*Improved estimation of ice COF from polarimetric phase co-registration'* or `*Improved estimation of ice COF from optimization of the polarimetric coherence'*

L 4, L 25, etc – I would use the term `polarimetric radar' rather than `radar'.

L 16 – is a new paragraph needed?

L 25 – Maybe `dielectric anisotropy due to crystal anisotropy'?

L 34 – A better description of what is meant by the `polarimetric coherence method' is needed here (something like: polarimetric coherence refers to the strength of the phase correlation between orthogonal polarizations)

L 35 – I have different interpretation of when the polarimetric coherence method will/will not work – see specific comments.

L 53 – should `accuracy ~ 1 mm ' be `precision ~ 1mm'?

L 56 onwards - I would be clearer from the offset that this study is considering muti-polarization data (i.e. co-polarized data as a function of azimuth), whereas most studies are now using quad-polarized data (also see specific comment 1).

L 71 – Does the East GRIP core contain azimuthal orientation and tilt measurements/zenith angle for the fabric eigenvectors? (Also see specific comment 2).

L 91 - I would replace `According to Fujita 2006 and Jordan 2019…' with `If it is assumed that the ice crystals are an effective medium at IPR frequencies…'

L 125 – I think a bit more context on the `fine-scale' ranging capabilities of ApRES is needed here. E.g. what is the physical interpretation of the $l/B_p$ term in equation (10)?

L 136 – I would quote the difference with the core data after this sentence

L158 – The use of quad-polarized data was proposed a long time before Ershadi et al. 2021. Notably, the work of Doake et al. 2003: https://folk.uib.no/ngfso/FRISP/Rep14/doake.pdf. Quad-polarized acquisition has the key advantage of reconstructing co-polarized data at a high angular resolution. I therefore think that combining the authors' co-registration method with the quad-pol method (e.g. Brisbourne et al. 2019, Young et al. 2021, Ershadi et al. 2021, Jordan et al. 2022), will be what people will do in the future, so I recommend writing a paragraph along these lines.

L176 onwards – As noted in specific comment 1, I think the comparison with Ershadi/other previous methods needs to be explicitly demonstrated to the reader.

---

## Referee Comment (RC2)

**Review of Zeising et al., TC, 2022**

The authors introduce a method for inferring the bulk horizontal anisotropy of glacier ice fabrics with depth from travel-time differences between radar waves with orthogonal polarizations.
The time-lagged cross-correlation method between the two waves represents an improvement over previous methods, which are discussed, and the case-study comparison with the (existing) EGRIP ice-core fabric profile is very convincing.
The structure and figures of the manuscript is/are well chosen, and I believe the advancements made will be received with great interest in the glaciological community.
In the end I have only minor comments in addition to the major issues #1 and #2 already raised by reviewer T. Jordan, which you may consider as you prepare the final manuscript.

Kind regards,

Nicholas Rathmann,
Niels Bohr Institute, UCPH, Denmark

**Minor comments:**

L1: I think it should be "ice c-axis fabric" (might be wrong).

L13: I would suggest "deformation history that can influence".

L14: I would delete the comma after "magnitude".

L17: I would replace "obstructed" with "challenged".

L21: I would add a comma after "Antarctica".

L23: Maybe mention again here that "[…] for improving ice-flow models and determining past flow/deformation".

L24-25: I would suggest rephrasing this slightly to something along:
"This means that ice crystals are dielectrically anisotropic, in addition to being mechanically anisotropic, and thus allow the horizontal fabric asymmetry to be determined from radar surveys […]".

L28: I would replace "achieve" with "conduct", and "good" with "greater".

L36: I would delete "severely".

L54: I would suggest "ice fabric from polarimetric measurements".

L60: More specifically I suppose you mean Fig. 1c .

L71: Not exactly clear what this means. Maybe consider reformulating this sentence?

L73: Would replace "c-axis" with "samples c-axes", and "by a second order" with "by the second-order".

L75: Would replace "correspond to the length of the three principal axes" with "quantifies the strength of the three principal fabric (c-axis) directions". Would also replace "derive" with "determine".

Eqn. 2, 4, 6, 7, 8, 10, Fig. 3, and other in-text occurrences:
While I appreciate the notational rigor, I think you could benefit (readability-wise) from dropping the *x'y'*

subscripts in *\delta t, \delta \epsilon, and \delta \lambda* (since you are only considering horizontal anomalies in this work anyway). Your single-crystal dielectric anisotropy could then be *\Delta \epsilon_c* (or some other subscript).

L85: Would replace "of the corresponding" with "in the corresponding".

Eqn. 5 and 6: I think you need to unfold this a bit more for the reader. How do these equations come about?

L90: "bulk" horizontal anisotropy.

Eqn. 7: Maybe note that this assumes wave lengths much longer than the average grain size.
Also, for context, I think it is worth mentioning (possibly elsewhere) that the eigenvalues represent only the strength of the *coarsest* degree of fabric anisotropy, and that finer fabric structure may exist although it cannot necessarily be detected with polarimetric radar (e.g. Hargreaves, 1978, or Rathmann et al., 2022).

L94: Do you mean to say this value applies for radar frequencies similar to those used by you? It can differ quite a bit (Fujita et al., 2000).

L100: Would add commas around "and advantage of".

L100-101:I am not entirely sure how to understand this. I would suggest you to rephrase it a bit.

L104-105: Please define what $s_{ij}$ is already here for the reader less familiar with the radio-glaciological nomenclature.

Eqn. 9: I find the summation limits a bit confusing. Normally, summation variables are indices (e.g. j=1,2,3, …), but you seem to mix it with the (discrete) depth variable, e.g. the upper limit $z_n + N$ is adding two quantities with different units?

L128: Would replace "we adopted" with "we changed".

L149: Would it be more accurate to replace "vertical distribution" with "vertical profile"?

L150: What uncertainty, precisely?

L161: Would delete "the" before "previous methods".

L169: Would delete "the" before "previous coherence method".

L179: Do you mean to say that the inverse method cannot handle such cases of rotation?

L180: Would re-phrase this more carefully as e.g. "[…] was unsuccessful and is another reason why we regard our method as an improvement […]".

L186: "at much greater depth".

L194: As this stands, I'm not sure it is sufficiently clear why this is the case. Maybe you could elaborate slightly.

---

## Author Comment (AC1)

**Authors point-to-point response on Referee Comment #1 to tc-2022-200**

**Please find our answers below the original comments from the Reviewer.**

**Review to Zeising et al. 2022, The Cryosphere**

**Summary**
Zeising et al. present a technical development using phase-sensitive radar polarimetry to estimate ice crystal orientation fabric near the EastGRIP ice core. The key point of novelty is to enhance the polarimetric coherence (a method used by previous radar polarimetry studies) by adjusting the range- bin offset between orthogonal polarizations. This allows them to obtain high coherence throughout the ice column, which is then used to infer two-way travel time differences, dielectric anisotropy, and azimuthal fabric anisotropy. They then compare with the ice core fabric eigenvalue differences, showing very good agreement down to ice depths ~ 1500 m.

I like the general concept, and enjoyed reading the paper. Whilst the coherence optimization/ range-bin offset method is a relatively simple refinement from previous studies, it produces agreement between ice-core fabric data and the radar asymmetry estimates (probably the best I have seen to date). However, as it stands, I think the paper needs to better demonstrate how it has improved coherence and fabric estimation from methods used in previous studies. Additionally, fabric orientation information should be provided (both from the radar, and comparison with the ice core if it is available) as other similar radar polarimetry studies have all done this. I also do not agree with the physical interpretation related to the `half-wavelength limit' (regarding why previous applications of polarimetric coherence will be ineffective) and have given some counter arguments and suggestions to rephrase the discussion.

At < 200 lines and 3 figures the study is significantly shorter than a typical TC article, and closer to what I would expect for a `TC brief communication'. I see no issue with this, but potentially the editorial team and authors will want to change the format for the final publication.

Best regards,
Tom Jordan, Plymouth Marine Laboratory, UK

**Specific/major comments**

*1. Demonstration of improved coherence and fabric estimates over previous methods.*

A central weakness of the study is that it does not explicitly demonstrate the improvement of the coherence magnitude and fabric estimates over previous methods. A reader less familiar with the field will therefore be uncertain about the progress made in the paper.
I think this can be fixed relatively easily by showing:

(i)  A depth profile for |chvv| calculated *without* the range-bin offset, similar to the second column in Fig 2. My guess is that this will decay rapidly with depth, showing the coherence method in the paper to be more effective than previous.
(ii) Adding fabric asymmetry estimates, where possible, using the previous approach: i.e. using vertical gradient of the co-polarized phase difference (no offset) employed by Dall 2010, Jordan et al. 2019, 2021, 2022, Young et al. 2021.

Related to this point, I would put a qualifying statement that the study can only be directly compared with methods that have used multi-polarized data (e.g. Jordan et al. 2019, 2020) rather than quad-pol approaches (e.g. Brisbourne et al. 2019, Young et al. 2021, Ershadi et al. 2021, Jordan et al. 2022). A consequence of using quad-pol is that it enables reconstruction of multi-polarization data at higher angular resolution, which gives a particular advantage to inferring fabric orientation.

Thanks for raising this point. We are happy to demonstrate the improvement of the polarimetric cross-correlation approach over the previous coherence method. The updated figure 3 in the manuscript (figure 1 below) shows in panel (a) the result of the improved method in comparison with the previous coherence approach by using the code from Young et al. (2021).

The previous approach from Young et al., 2021 which is using the vertical gradient of the co-polarised phase difference, shows a similar vertical profile of the fabric asymmetry between 100 and within the upper 260 m. Below, the asymmetry drops and stays close to zero to a depth of 1000 m. Thus, the new cross-correlation approach is able to retrieve the fabric asymmetry over 5 times larger depth.

This comparison demonstrates the significant improvement over the previous methods in case of a strong fabric asymmetry as it is the case at the EastGRIP drill site.

We decided not to show the depth profile obtained from the new method but without the range-bin offset. In theory, it would be possible to follow the same phase-shift minimum to a depth of 250 m, but the 200 m-window over which the slope is calculated, prevents reliable results between 150 - 250 m.

We agree that one advantage of quad-pol data is to infer the fabric orientation from synthesising the data set. We will make this point clear in the manuscript.

[Figure]

**Figure 1:** Comparison of horizontal fabric asymmetry determined from different measurements and analysing methods. (a) Fabric asymmetry determined from cross-correlation analysis (lines) of pRES measurements at CL (light blue line) and at the 20×20 m GRID outside drill site as well as from the previous coherence method (Young and Dawson, 2021) at CL (light blue dots). (b) Fabric asymmetry determined from cross-correlation analysis (lines) of pRES measurements at CL (light blue line) and from weighted horizontal eigenvalues from EastGRIP ice core (black and white dots). The blue shaded area in (b) marks the range of the polarimetric pRES-derived asymmetry from the measurements in the GRID and at CL.

*2. Fabric orientation information.*

In addition to azimuthal asymmetry, polarimetric radar enables estimation of the orientation of (assumed) horizontal eigenvectors. This can be done by comparing data with the polarimetric backscatter model (Fujita 2006, Jordan et al. 2019) and using sign of the coherence phase gradient (Dall 2010). Fabric orientation is very useful information, as it informs about past deformation, aswell as being key information for understanding the impact of fabric on anisotropic rheology.

I therefore find the current study incomplete, and I think it would be significantly strengthened, by including fabric orientation panels in Fig 3 (i.e. azimuthal angle of horizontal eigenvectors as a function of ice depth). I appreciate that the angular resolution will be limited as multi-polarization rather than quad-pol was used, but I still think this will be a nice inclusion.

Additionally, does the EastGRIP core have azimuthal fabric orientation to compare with? If it does, then this should be included in the orientation plots.

> The presented study aimed to describe a new method that significantly improves the determination of the fabric asymmetry from polarimetric radar measurements. Here, we used a data set from the EastGRIP drill site as an example, as it is the first deep ice core with analysed fabric eigenvalues that was drilled into an ice stream. We fully understand the demand of the fabric orientation at the EastGRIP drill site. However, we would like to address this in an additional manuscript in which we will analyse the spatial variability of the ice fabric (strength of the asymmetry and the orientation) in the NEGIS-region around EastGRIP.

*3. Accuracy of the discussion on the `half-wavelength limit'*

In their discussion, the authors focus on the `half-wavelength limit' (for the polarimetric phase difference) as an explanation why the polarimetric coherence method has previously been limited. e.g. `Due to the ambiguities caused by phase wrapping, the previous methods which are based on the coherence phase gradient were limited to the derivation of phase shifts of a maximum of half a wavelength, (Line 61)'. I don't agree with this interpretation, and I have given some counter arguments and suggestions for revision below.

First, I don't think previous methods based on phase gradient are limited by the phase wrapping. Jordan et al. 2019 used an identity (equation 23) to differentiate the phase. This approach (adapted from the InSAR literature) gets around the issue of phase wrapping, as the real and imaginary components of the coherence are continuous functions, enabling the derivative to be taken. Figure 5 and 6 from this paper illustrates that the phase gradient can be obtained at the phase discontinuities.

There are other examples in the literature that illustrate that the coherence magnitude, phase difference, and fabric estimates are not physically limited by the phase discontinuities and the proposed `half-wavelength limit'. Notably, Young et al. 2021 (Fig. 4) shows high coherence persisting over multiple (~4) phase cycles.

It therefore follows that `strong azimuthal asymmetry' (and rapid phase-cycling) should not be a singular limitation on the previous method. I think it is probably coincidence that the fabric in Jordan et al. 2022 is often only obtained in the first phase cycle (line 171). This study proposed a degradation in the radar stratigraphy as the reason for the coherence drop-off with depth. Despite discounting the `half-wavelength limit' interpretation, I do agree with the authors that their optimization should lead to higher coherence, and therefore improve the fabric estimates. This is because their method should act to better co-register the signal from a given radar layer. I think, if they better show the impact of the optimization on the coherence (following my comment 1), then they will be able to refocus the discussion around improved co-registration being the physical mechanism that improves the coherence and fabric estimates. (As an aside, I don't think it is strictly necessary for the reflection to occur from the same layer for each polarization. As long as the layers behave as flat, specular, reflectors, the original coherence method should still work to some degree).

As well as the discussion, lines 35-38 and line 186, will also need addressing regarding this point.

Thanks for raising this point. We tried to understand the raised arguments and we agree that it can be possible to analyse the phase gradient from the polarimetric phase difference that exceeds half a wavelength in some cases. However, we argue that this is not possible in general and all cases.

The strong fabric asymmetry at EastGRIP causes a rapid phase-cycling and a reduction in coherence after a few hundred meters. Already the third phase cycle shows a different phase gradient and low coherence. We attribute this to the fact that, due to the difference in propagation velocity, the correlated segments do not overlap sufficiently. We see this as the main limitation of the coherence method for strongly developed asymmetries that we can overcome with the co-registration by the cross-correlation.

In the revised version of this manuscript, we will rewrite this part of the discussion.

However, we prefer to keep the sentences on lines 35-38 and 186 (previous version) as we consider them correct. We are aware that Jordan et al. 2022 proposed that a degradation in the radar stratigraphy is the reason for the reduced coherence. However, the application of the polarimetric cross-correlation method to their measurements showed that this limit can be overcome, which speaks against this explanation.

The new part of the discussion will be:

*"The previously used coherence method estimates the fabric asymmetry by determining the phase gradient of the polarimetric phase difference. This is also possible for high coherence persisting over a few phase cycles (e.g., Young et al., 2021a). However, in case of a strongly developed fabric asymmetry and thus a rapid phase-cycling, the coherence is reduced over depth because the segments that are correlated do not completely overlap and therefore contain different scatterers (Leinss et al., 2016). At ice divides or domes with very little asymmetry, such as at NEEM (Jordan et al., 2019), WAIS divide (Young et al., 2021a) or EDC (Ershadi et al., 2022), the fabric asymmetry could successfully be determined using previous coherence method up to the onset of noise. However, in fast moving areas like the Rutford Ice Stream, Antarctica (Jordan et al., 2022) or NEGIS, Greenland (this study) rapid phase-cycling limits the application of the previous coherence method to a few hundred meters below the surface. With the improved polarimetric cross-correlation method, we overcome this limitation through co-registration, which allows to determine even strong horizontal fabric asymmetries to a much greater depth."*

**Minor comments/typos**

Title - The MS title is quite generic (in effect all COF/radar polarimetry studies estimate horizontal asymmetry!). As the phase co-registration/optimization of the coherence is the key point of novelty, I recommended changing to something like: *`Improved estimation of ice COF from polarimetric phase co- registration'* or *`Improved estimation of ice COF from optimization of the polarimetric coherence'*

We are grateful for your title suggestions and will change the title to:

*"Improved estimation of the bulk ice crystal fabric asymmetry from polarimetric phase co-registration"*

L 4, L 25, etc – I would use the term `polarimetric radar' rather than `radar'.

Thanks, we will add "polarimetric" to be more precise.

L 16 – is a new paragraph needed?

We will remove the line break.

L 25 – Maybe `dielectric anisotropy due to crystal anisotropy'?

Thanks! We will change this sentence accordingly.

L 34 – A better description of what is meant by the `polarimetric coherence method' is needed here (something like: polarimetric coherence refers to the strength of the phase correlation between orthogonal polarizations)

Thanks, we will implement such a description in the revised version.

L 35 – I have different interpretation of when the polarimetric coherence method will/will not work – see specific comments.

Please see our answer to the specific/major comment.

L 53 – should `accuracy ~ 1 mm ' be `precision ~ 1mm'?

Yes, thanks. Will will change this in the revised version.

L 56 onwards - I would be clearer from the offset that this study is considering muti-polarization data (i.e. co-polarized data as a function of azimuth), whereas most studies are now using quad-polarized data (also see specific comment 1).

We will make clear that this study uses co-polarised measurements and add the following sentences in the data acquisition section:

*"Recent studies used quad-polarised acquisitions which additionally include hv and vh measurements, where the polarisation direction of the transmitting and receiving antenna is rotated by 90° (e.g. Brisbourne et al., 2019; Young et al., 2021a; Ershadi et al., 2022; Jordan et al., 2022). However, this study focuses on co-polarised measurements."*

In the discussion section, we will also mention the advantages of quad-polarised measurements explicitly (please see comment further below).

L 71 – Does the EastGRIP core contain azimuthal orientation and tilt measurements/zenith angle for the fabric eigenvectors? (Also see specific comment 2).

Not yet. Deriving the eigenvectors is work in progress and will be published in a dedicated study on the crystal orientation within the EastGRIP ice core.

L 91 - I would replace `According to Fujita 2006 and Jordan 2019...' with `If it is assumed that the ice crystals are an effective medium at IPR frequencies…'

Thanks, we will change this sentence accordingly.

L 125 – I think a bit more context on the `fine-scale' ranging capabilities of ApRES is needed here. E.g. what is the physical interpretation of the l/Bp term in equation (10)?

Thanks. We will add an additional description like:

*"The first term on the right side is the coarse time-bin offset with 1/Bp being the time-bin spacing (B = 200 MHz is the bandwidth and p = 8 is a 'padding factor' that reduces the range-bin spacing). The second term is the fine offset derived from the coherent phase of the centre frequency of f_c = 300 MHz."*

L 136 – I would quote the difference with the core data after this sentence

Done.

L158 – The use of quad-polarized data was proposed a long time before Ershadi et al. 2021. Notably, the work of Doake et al. 2003: https://folk.uib.no/ngfso/FRISP/Rep14/doake.pdf. Quad-polarized acquisition has the key advantage of reconstructing co-polarized data at a high angular resolution. I therefore think that combining the authors' co-registration method with the quad-pol method (e.g. Brisbourne et al. 2019, Young et al. 2021, Ershadi et al. 2021, Jordan et al. 2022), will be what people will do in the future, so I recommend writing a paragraph along these lines.

We agree that a quad-polarised acquisition has a clear advantage against co-polarised measurements at different azimuthal angles. We have tested the application of the new cross-correlation method to reconstructed co-polarised data from quad-polarised measurements and came to the same vertical profile of the fabric asymmetry.

We will add the following paragraph to the discussion:

*"However, a clear advantage of quad-polarised measurements is that they allow to reconstruct co-polarised data at a high angular resolution and additionally the determination of the fabric orientation (e.g. Brisbourne et al., 2019; Young et al., 2021a; Ershadi et al., 2022; Jordan et al., 2022). The presented cross-correlation method can also be applied to these reconstructed co-polarised data. Since only four measurements (hh, vv, hv and vh) at one azimuthal angle are necessary to perform a quad-polarised acquisition, but eight for co-polarised measurement (hh and vv) at four different azimuthal angles (0°, 22.5°, 45° and 67.5°), quad-polarised measurements should be preferred in the future"*

L176 onwards – As noted in specific comment 1, I think the comparison with Ershadi/other previous methods needs to be explicitly demonstrated to the reader.

We are happy to compare the results of the previous methods to show the improvement of the new method. As mentioned above, we will do that on the example of the code from Young et al., 2021.

---

## Author Comment (AC2)

**Authors point-to-point response on Referee Comment #2 to tc-2022-200**

**Please find our answers below the original comments from the Reviewer.**

**Review of Zeising et al., TC, 2022**

The authors introduce a method for inferring the bulk horizontal anisotropy of glacier ice fabrics with depth from travel-time differences between radar waves with orthogonal polarizations. The time-lagged cross-correlation method between the two waves represents an improvement over previous methods, which are discussed, and the case-study comparison with the (existing) EGRIP ice-core fabric profile is very convincing.

The structure and figures of the manuscript is/are well chosen, and I believe the advancements made will be received with great interest in the glaciological community.
In the end I have only minor comments in addition to the major issues #1 and #2 already raised by reviewer T. Jordan, which you may consider as you prepare the final manuscript.

Kind regards,
Nicholas Rathmann,
Niels Bohr Institute, UCPH, Denmark

**Minor comments:**

L1: I think it should be "ice c-axis fabric" (might be wrong).

> We think both ways are fine and we decided to keep "c-axes" as we are talking about the sum of all samples.

L13: I would suggest "deformation history that can influence".

> Thanks, we will add this in the revised version.

L14: I would delete the comma after "magnitude".

> Done.

L17: I would replace "obstructed" with "challenged".

> Thanks, we will do so.

L21: I would add a comma after "Antarctica".

> We will add a comma.

L23: Maybe mention again here that "[...] for improving ice-flow models and determining past flow/deformation".

> Thanks, we will change this sentence accordingly.

L24-25: I would suggest rephrasing this slightly to something along:
"This means that ice crystals are dielectrically anisotropic, in addition to being mechanically anisotropic, and thus allow the horizontal fabric asymmetry to be determined from radar surveys [...]".

> Thanks for your suggestion. Also Referee #1 suggested a reformulation. We will change the sentence follows:

*"This means that ice crystals are dielectrically anisotropic due to crystal anisotropy and thus allow the horizontal fabric asymmetry to be determined from polarimetric radar surveys […]."*

L28: I would replace "achieve" with "conduct", and "good" with "greater".

Thanks. We will change both words in the revised version.

L36: I would delete "severely".

Done.

L54: I would suggest "ice fabric from polarimetric measurements".

We will change this sentence as suggested.

L60: More specifically I suppose you mean Fig. 1c.

Yes, thanks.

L71: Not exactly clear what this means. Maybe consider reformulating this sentence?

Thanks. We will rewrite this sentence as follows:

*"Every 5–15 m of depth of the ice core a 55 cm long section was analysed for fabric data."*

L73: Would replace "c-axis" with "samples c-axes", and "by a second order" with "by the second-order".

Thanks. We will change the sentence as follows:

*"The grain size weighted orientation of the measured c-axes can be represented by the second-order orientation tensor."*

L75: Would replace "correspond to the length of the three principal axes" with "quantifies the strength of the three principal fabric (c-axis) directions". Would also replace "derive" with "determine".

Done. Thanks!

Eqn. 2, 4, 6, 7, 8, 10, Fig. 3, and other in-text occurrences: While I appreciate the notational rigor, I think you could benefit (readability-wise) from dropping the x'y' subscripts in \delta t, \delta \epsilon, and \delta \lambda (since you are only considering horizontal anomalies in this work anyway). Your single-crystal dielectric anisotropy could then be \Delta \epsilon_c (or some other subscript).

We would prefer to keep the subscripts to avoid misunderstanding. Especially, \Delta \epsilon is something different than \Delta \epsilon_xy.

L85: Would replace "of the corresponding" with "in the corresponding".

Done.

Eqn. 5 and 6: I think you need to unfold this a bit more for the reader. How do these equations come about?

> We added new equations (eq. 2+3 in revised version)
>
> $$\overline{v}_{x'}(z) = \frac{c_0}{\sqrt{\overline{\varepsilon}_{x'}(z)}} = \frac{2z}{t_{x'}(z)},$$
>
> $$\overline{v}_{y'}(z) = \frac{c_0}{\sqrt{\overline{\varepsilon}_{y'}(z)}} = \frac{2z}{t_{y'}(z)}$$
>
> as the basic equations to introduce the eqs. 3+4 (previous version; 5+6 revised version) and additionally, we explaining the transition to eqs. 5+6 (previous version; 5+6 revised version):
>
> *"These dielectric permittivities are the average values over the entire depth from the surface to the depth z. In order to calculate the vertical profile of the horizontal dielectric anisotropy $\Delta\varepsilon\_x'y' = \varepsilon\_y' - \varepsilon\_x'$, the local change in two-way travel time $\delta t\_x'$, $\delta t\_y'$ for a given infinitesimal depth window $\delta z$, needs to be taken into account: [...]"*

L90: "bulk" horizontal anisotropy.

> Done.

Eqn. 7: Maybe note that this assumes wave lengths much longer than the average grain size.

> Done.

Also, for context, I think it is worth mentioning (possibly elsewhere) that the eigenvalues represent only the strength of the coarsest degree of fabric anisotropy, and that finer fabric structure may exist although it cannot necessarily be detected with polarimetric radar (e.g. Hargreaves, 1978, or Rathmann et al., 2022).

> In the introduction, we already mentioned that there are limitations and cited Rathmann et al. (2022). Thus, we think that it is not necessary to mention this more explicitly.

L94: Do you mean to say this value applies for radar frequencies similar to those used by you? It can differ quite a bit (Fujita et al., 2000).

> It is true that the dielectric anisotropy for a single crystal can differ from 0.034 depending on the frequency and the temperature. However, Matsuoka et al., 1997 found the same dielectric anisotropy for 1 MHz as well as for 39 GHz. Also all recent studies that analysed polarimetric pRES measurements used a dielectric anisotropy of 0.034.

L100: Would add commas around "and advantage of".

> Done.

L100-101:I am not entirely sure how to understand this. I would suggest you to rephrase it a bit.

> We will simplify this sentence by removing "difference to and". So the new sentence will be:
>
> *"This is the main advantage of the in-depth analysis of the phase which is why polarimetric pRES measurements offer the chance to investigate the horizontal fabric asymmetry in the ice."*

L104-105: Please define what s_ij is already here for the reader less familiar with the radio-glaciological nomenclature.

We will add the following explanation:

*"(subscripts indicate the transmitted and received polarisation)"*

Eqn. 9: I find the summation limits a bit confusing. Normally, summation variables are indices (e.g. j=1,2,3, ...), but you seem to mix it with the (discrete) depth variable, e.g. the upper limit z_n + N is adding two quantities with different units?

The summation variable is actually an indices: z_n is the range bin index of the beginning of the segment. Thus, z_n has no dimension. The segment itself consists of N values. Thus, the summation limit is z_n+N.

For more clearness, we will change z_n to i_n in the revised version.

L128: Would replace "we adopted" with "we changed".

Done.

L149: Would it be more accurate to replace "vertical distribution" with "vertical profile"?

Yes, we will do so. Thanks!

L150: What uncertainty, precisely?

Here, we actually meant the scatter of Delta t_xy due to the uncertainty. We will correct this:

*"Despite the high range resolution, the scatter of Δt_x'y' caused by the uncertainty prevented a determination of the small-scale gradient of the travel-time difference."*

L161: Would delete "the" before "previous methods".

Done.

L169: Would delete "the" before "previous coherence method".

Done.

L179: Do you mean to say that the inverse method cannot handle such cases of rotation?

Yes. We will reformulate this as follows to make it clearer:

*"However, the ice fabric orientation in this area rotates several times at different depths of the ice column, which prevents the application of the previous method using the inverse approach."*

L180: Would re-phrase this more carefully as e.g. "[...] was unsuccessful and is another reason why we regard our method as an improvement [...]".

Thanks! We will follow your suggestion.

L186: "at much greater depth".

Thanks, we will change this accordingly.

L194: As this stands, I'm not sure it is sufficiently clear why this is the case. Maybe you could elaborate slightly.

We reformulated this part as follows:

*"Such an application, which would yield the variation of the horizontal anisotropy along flow lines or across regions of fast flow, like ice streams, would significantly improve the understanding of the link between the stress state and crystal fabric evolution. This would allow us to decrease uncertainties of rheology, and thus improve estimates for response times of dynamically active glacial systems to external perturbations, e.g. from changing ocean conditions of tidewater glaciers."*

---

## Author Response (AR1)

**Authors point-to-point response on Editor Comment #1 to tc-2022-200**

Dear Editor, dear Joe MacGregor,

We thank you for your two important and helpful comments. Please find our answers below in green.

1. *The statement of "an almost perfect agreement" in the abstract yet the lack of a direct quantification of the relation between the radar and ice-core fabric measurements is not satisfactory. This needs to be quantified, whether by a simple correlation coefficient, rms difference or some other suitable metric. I recognize that the relevant eigenvalue difference varies with depth, so perhaps a depth subset is appropriate/simpler, but the present situation is dissonant for what is an other quantitatively rigorous MS.*

> Thanks for raising this point. We had a short statement about the mean difference in the results section. Now, we calculated the root-mean-square of the differences of both methods (radar and ice core analysis) by one subset between 120 m and 250 m depth for the difference of the Eigenvalues 1 & 2 and and a second subset below for the difference of the Eigenvalues 1 & 3.

> We improved and moved the following section to the discussion (l. 166 - 171):

> *"The horizontal fabric asymmetry derived from the polarimetric cross-correlation of the pRES measurements and the difference of the weighted horizontal eigenvalues from the ice core analysis ($\lambda 2$–$\lambda 1$ between 120 and 250 m and $\lambda 3$–$\lambda 1$ between 250 and 1400 m) show almost perfect agreement with a root-mean-square difference of the result of both methods of only 0.03. This value corresponds to the uncertainty of the ice core analysis and thus represents the lowest possible value in the difference. However, the root-mean-square value of the difference of the unweighted horizontal eigenvalue is 0.06 and thus higher, which is a result compatible to analyses of seismic waves by Kerch et al. (2018)."*

2. *What is the ice thickness at both the study sites, and for what fraction of the ice column is the S/N high enough to analysis? There are several paths here. First, the ice thicknesses ought to be given in the text. Second, in Figures 2 and 3 either show the complete radar dataset down to the bed (or the noise floor, whichever is first), or show an additional vertical axis that is the fraction of the ice thickness rather than absolute depth. Otherwise, the casual reader could be left with the impression that you've measured fabric through the entire ice column, rather than what I suspect is closer to half of it. Still a big improvement over earlier methods, but a more sober representation of the outcome. Related to this, it could be worth discussing how we might eventually be able to detect bulk fabric changes all the way to bed.*

> You are absolutely right, we have missed stating the ice thickness, which is roughly 2668m at the EastGRIP drill site. In the revised version, we state this in the Data section. We added a second y-axis to Fig. 3 with the fraction of ice thickness and mentioned these fractions for certain depths in the result section. We also added the following section to the discussion (l. 192 – 197):

> *"Noise limits the evaluation of fabric asymmetry for deeper layers. At the EastGRIP drill site, this limit is about half the ice thickness of the ice with current systems. Determining the fabric for deeper layers from radar measurements, eventually over the whole ice sheet thickness, requires further reduction of the signal-to-noise ratio in more powerful phase-sensitive radar system that can perform co- or quad-polarized measurements. The applicability of the polarimetric cross-correlation method needs first to be demonstrated for such radar systems."*

We have improved the manuscript according to the responses to the reviewers.

Many thanks for your efforts to improve our manuscript!

Best regards,

Ole Zeising

---

## Author Response (AR2)

**Authors point-to-point response on Editor Comment #2 to tc-2022-200**

Dear Editor, dear Joe MacGregor,

Thanks for your quick responds and the new suggestions. We have made all three changes accordingly.

Many thanks for your efforts to improve our manuscript!

Best regards,

Ole Zeising